

# Interlaboratory comparison exercise for micro-aerosol size measurement by cascade impactor

Grégoire Dougniaux[1], Cécile Bodiot[1], Bernadette Dhieux Lestaevel[1], Amandine Nuboer[2], Robin Wahl[2], Amel Kort[1]

[1]Institut de Radioprotection et de Sûreté Nucléaire (IRSN), PSN-RES/SCA/LPMA, F-91400, Saclay, France
[2]Institut de Radioprotection et de Sûreté Nucléaire (IRSN), PSN-RES/SCA/LECEV, F-91400, Saclay, France

*Correspondence to*: Grégoire Dougniaux (gregoire.dougniaux@irsn.fr)

**Abstract.** This study presents an interlaboratory comparison (ILC) exercise focused on measuring micro-aerosol size distributions using cascade impactors. The aerodynamic particle size distribution (APSD) is a critical parameter for understanding aerosol behaviour, particularly for health-related applications. The ILC conducted at the Institut de Radioprotection et de Sûreté Nucléaire (IRSN) aims to assess the performances of participating instruments measuring aerodynamic diameter, cascade impactors and an Aerodynamic Particle Sizer (APS) for real time monitoring. The experiments were performed in a custom test bench able to generate aerosols in a size range from 0.2 to 4 µm within a controlled environment. Performance evaluations of the participating instruments considering five distinct aerosol size distributions were assessed, and two methods – Henry's method and lognormal adjustment – were used to calculate the mass median aerodynamic diameter (MMAD) and the geometric standard deviation ($\sigma_g$). Statistical analysis using $\zeta$-score and Z'-score ensured the reliability of the results across participating instruments.

The findings demonstrates that most instruments performed within acceptable limits, though variations observed in some cases, particularly for smaller particle sizes. This work highlights the feasibility of standardized ILCs for APSD measurement and offers a framework for improving accuracy and consistency in aerosol size distribution assessments.

## 1 Introduction

Particle size-distribution is a key parameter to describe particle behaviour – formation, transport, fate…. In terms of health, the 6[th] leading cause of death worldwide is the tracheal, bronchial and lung cancer (WHO, 2024). These kinds of diseases are related to the effects of aerosol inhalation. Manisalidis et al. (2020) review the environmental and health effects of air pollution consequences on morbidity and mortality. Air pollution is almost systematically caused by human activities, and composed of different particles, such as Volatile Organic Compounds (VOCs), Polycyclic Aromatic Hydrocarbons (PAHs), Particulate Matter (PM), … Increasing attention is being paid to the health consequences of PM emitted by different processes.

The aerosol behaviour in the respiratory system is standardized with sampling conventions (ICRP, 1994, 2002; NF EN 481, 1993) that uses the same measurand: the aerodynamic diameter. The Aerodynamic Particle Size Distribution (APSD) is associated to the effects of particle deposition in the human respiratory system (Heyder et al., 1980).



APSD can be measured using real time instruments such as Aerodynamic Particle Sizer, Electrical Low-Pressure Impactor, Quartz Crystal Micro Orifice Uniform Deposit Impactor… and differing time instruments such as cascade impactors or Low-Pressure Impactor … The multi-stage inertial cascade impactor is considered as the "gold standard" for aerodynamic particle size distribution assessment.

However, there is currently no calibration chain linked to the international system of units for aerodynamic diameter. Proficiency Testing (PT) is widely recognised as an essential tool for demonstrating the competence of conformity assessment organisations. PTs can provide evidence of competence and involve the use of Inter-Laboratory Comparisons (ILC) to assess laboratory performance.

Few ILCs have been performed on the APSD. Pfeifer et al. (2016) set-up an ILC on fifteen Aerodynamic Particle Sizer (APS),
and no cascade impactor. In 2016, Fonseca et al. proposed an ILC on four different cascade impactors using a BLPI as reference and a natural ambient aerosol of two European cities. In 2020, Gaie-Levrel et al., after performing an ILC on Scanning Mobility Particle Sizers (SMPS), proposed to continue with an ILC on APSD.

Therefore, we have developed a test bench at IRSN to perform an ILC on APSD over a wide range of particle sizes. In this article, we describe the feasibility of the ILC on this test bench as well as the results involving cascade impactors (Andersen
Cascade Impactor, Dekati Low Pressure Impactor (DLPI), DLPI+) and an Aerodynamic Particle Sizer (APS) for real time monitoring. This first ILC on APSD measurement follows the requirements and methodology of the standards for PT organisers (ISO/IEC 17043, 2023) and the statistical methods for PT with ILC (Amarouche, 2015; ISO 13528, 2022).

## 2 Experimental setup

The enclosed test bench (Figure 1), used to perform the ILC is designed to handle aerosols in complete safety without
interference from the background noise of the surrounding atmospheric aerosol. This test bench was set up at IRSN to generate and study particles with diameter down to nanometric range (Brochot et al., 2012). The test bench consists of a 0.48 m³ test chamber with a controllable closed-circuit airflow from 100 to 300 m³/h. The test bench can operate with up to 100% of recycled air using a High Efficiency Particulate Air (HEPA) filtration system. An aerosol generator is connected to the bench to produce the expected particle concentration in the measuring chamber. The aerosol is sampled by the ILC instruments
through calm air probes (Grinshpun et al., 1993, 1994) to the test instruments.



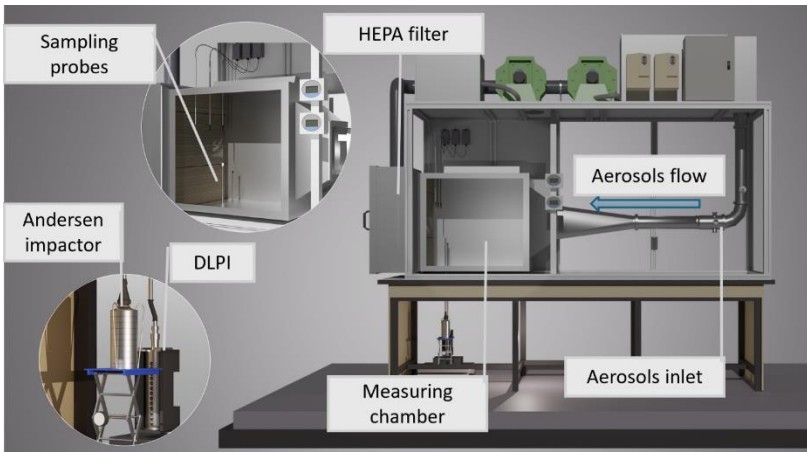

**Figure 1 – Schematic of the test bench**

Five aerosols with target aerodynamic diameters between 0.2 and 4 µm were selected for this ILC. Table 1 describes the characteristics of the solutions and generators used to produce these aerosols. The solutions are made of $H_2O$, KCl and sodium fluorescein.

**Table 1 – Five aerosols generation characteristics**

| Test ID | Aerosol material | KCl concentration g/L | Fluorescein concentration mg/L | Aerosol generator |
|---------|------------------|-----------------------|--------------------------------|-------------------|
| A | | 300 | 12 | Nebulizer SINAPTEC 1 MHz |
| B | | 200 | 15 | |
| C | KCl | 40 | 15 | |
| D | | 8 | 6 | Atomizer TSI 3076 |
| E | | 0,1 | 15 | |

Table 2 shows all the instruments used for the ILC. It should be noted that real-time monitoring of the particle size distribution is provided by an APS for aerosols larger than 1 µm.



**Table 2 – ILC participants**

| Participant ID | Instrument | Measurand | Range |
|:---:|:---:|:---:|:---:|
| 1 | Impactor Andersen Mark II (Tisch) | | 0,4 – 9 µm 8 channels |
| 2 | DLPI+ (Dekati) | Aerodynamic diameter measurement by mass | 0,006 – 10 µm 15 channels |
| 3 | DLPI (Dekati) | | 0,03 – 10 µm 13 channels |
| 4 | APS (TSI 3021) | Aerodynamic diameter measurement by time of flight | 0,5 – 20 µm |

### 3 Analysis methods

For all the instruments, the two calculation methods used to obtain the Mass Median Aerodynamic Diameter (MMAD) and geometric standard deviation ($\sigma_g$) are Henry's method and lognormal adjustment. The result of the real-time monitoring instrument is also given, as well as the target diameter derived from knowledge of the generators. The log-normal distribution is commonly used in aerosol analysis because it can model particle size distributions over a wide range of diameters, accounting for the typically skewed nature of such distributions.

The Henry's method, also known as the Q-Q plot (quantile-quantile plot), is a tool used to compare observed data distributions to a theoretical distribution, in this case a log-normal distribution. By plotting theoretical quantiles against measured ones, Henry's method enables the determination of MMAD and $\sigma_g$ with well quantified uncertainties.

The log-normal adjustment assumes that the size distribution of aerosol particles follows a log-normal distribution. This method involves fitting the experimental data to a log-normal law, characterized by two main parameters: the MMAD and the
$\sigma_g$ and their uncertainties.

To obtain the aerosol mass on each impactor filter, thus the APSD, a fluorometric analysis is performed. For each impactor, all the filters are taken then put individually in an ammoniac solution (pH>10) to solubilize the fluorescein. The latter is quantified with a calibrated fluorimeter. This calibration and the known relation between the KCl and fluorescein concentrations (Table 1) allow to establish the aerosol mass sampled on each filter.

The aim is to present the deviation from the reference value in a way that allows simple and consistent interpretation between different ILC campaigns and different measurands properties. The reference value is calculated from the results of the three impactors – the APS is used as a monitoring instrument. The global uncertainties are calculated from the dispersion of the three repetitions and the uncertainties of the methods used to calculate the parameters (Henry's method and log-normal adjustment). Two indicators are used: the $\zeta$-score and the Z'-score (equations 1 and 2).


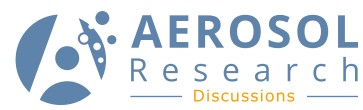

$$\zeta = \frac{x_p - x_{ref}}{\sqrt{u_p^2 + u_{ref}^2}}$$

(**1**)

$$Z' = \frac{x_p - x_{ref}}{\sqrt{\sigma_p^2 + u_{ref}^2}}$$

(**2**)

where:

- $x_p$: measurement result
- $x_{ref}$: reference value
- $u_p$: standard uncertainty associated with the measurement result
- $u_{ref}$: standard uncertainty associated with the reference value
- $\sigma_p$: robust standard deviation of all measurement results

The conventional interpretation of the indicators is as follows:

- $|Z'| \leq 2$        $|\zeta| \leq 2$            acceptable results
- $2 < |Z'| < 3$      $2 < |\zeta| < 3$          doubtful result, alert threshold
- $|Z'| \geq 3$        $|\zeta| \geq 3$            unacceptable results

The ζ-score is a deviation normalised by the uncertainties of the reference ($u_{ref}$) and the participant ($u_p$). A large absolute value indicates an underestimated experimental uncertainty.

The Z'-score is a deviation normalised by the uncertainties of the reference ($u_{ref}$) and the standard deviation of the participant ($\sigma_p$). A large absolute value indicates that the result is statistically incompatible with the reference value.

The reference value is established from all the participants' results using robust means and standard deviations (NF ISO 13528:2022 §C.3, Algorithm A).

## 4 Results and discussion

Each instrument is connected to the test bench and samples during 15 min. Then, the filters are exploited to obtain the deposited masses. This process is repeated three times. Finaly, table 3 presents the reference values for this ILC, figure 2 and figure 3 the Z'-score and ζ-score for each participant.

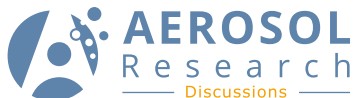

**Table 3 – Reference values for the ILC**

| Test ID | Reference value<br>$x_{ref}$ (µm) | Standard uncertainty<br>$u_{ref}$ (µm) | Standard deviation<br>$\sigma_p$ (µm) |
|---------|----------------|----------------------|--------------------|
| A | 3.947 | 0.061 | 0.24 |
| B | 1.817 | 0.027 | 0.11 |
| C | 0.951 | 0.029 | 0.098 |
| D | 0.549 | 0.015 | 0.052 |
| E | 0.3254 | 0.0050 | 0.014 |

The Z'-score and ζ-score calculated for this ILC are presented on figure 2 (MMAD) and figure 3 ($\sigma_g$) respectively on the left

and on the right, for all the participants (Table 2).

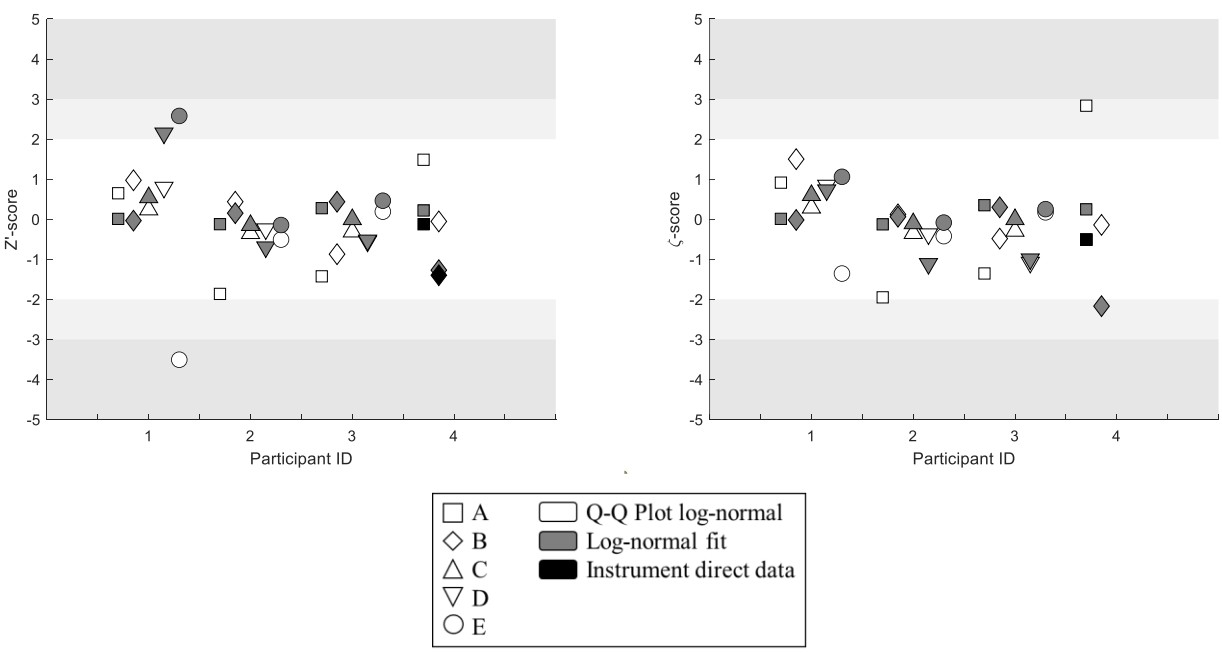

**Figure 2 – Z'-score (left) and ζ-score (right) calculated for the MMAD**






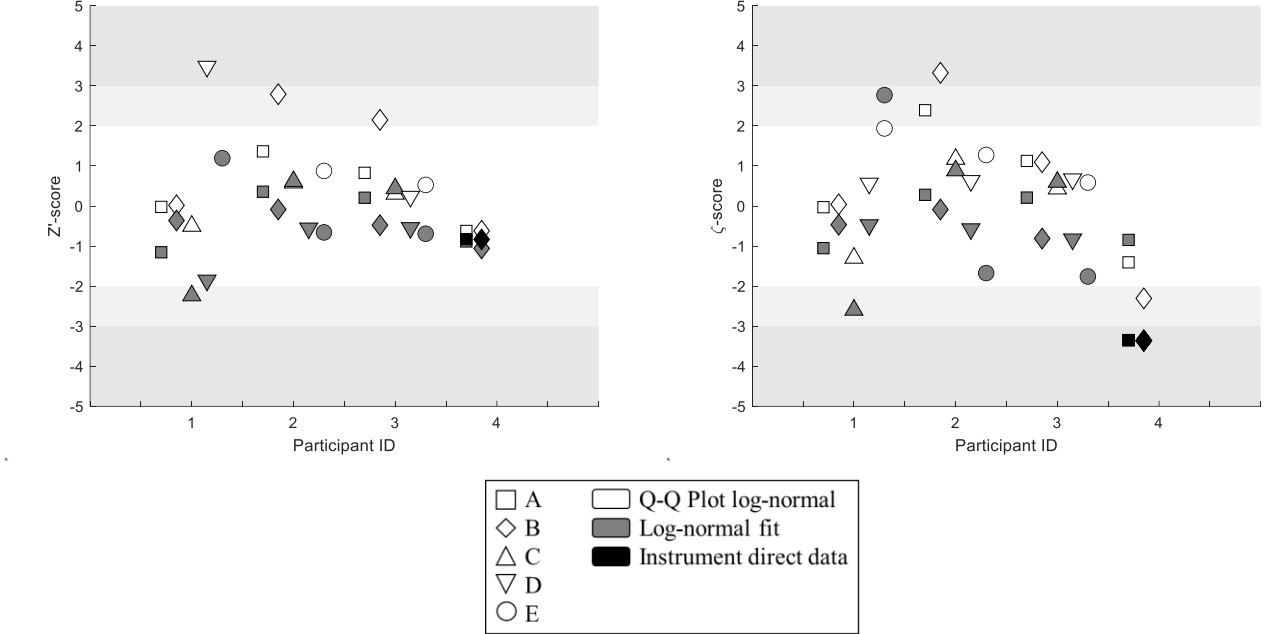

**Figure 3 – Z'-score (left) and ζ-score (right) calculated for the $\sigma_g$**

The scores for the MMAD (figure 2) and the $\sigma_g$ (figure 3) can be analysed this way:

1)    Participant #1 pass successfully the three first tests (A, B & C), for both analysis methods.

       However, it has difficulties to give reliable results below its range (tests D & E), which is expected.

2)    Participants #2 & #3 pass successfully the five tests, for both analysis methods.

3)    Participant #4 gives acceptable MMAD and $\sigma_g$ but tends to underestimate the experimental uncertainties.

4)    The Henry's method (Q-Q plot) gives more disperse results, seems to underestimate the experimental uncertainties.

**Conclusion**

This interlaboratory comparison (ILC) exercise demonstrated the feasibility and effectiveness of a standardized approach for assessing aerodynamic particle size distribution (APSD) using cascade impactors. To conduct the ILC, a specific test bench has been set-up able to generate aerosols in a size range from 0.2 to 4 µm within a controlled environment. Four IRSN

instruments have been used to demonstrate our ILC capability. Performance evaluations of the participating instruments considering five distinct aerosol size distributions were assessed, and two methods – Henry's method and lognormal adjustment – were used to calculate the mass median aerodynamic diameter (MMAD) and the geometric standard deviation ($\sigma_g$). Statistical analysis using ζ-score and Z'-score ensured the reliability of the results across participating instruments. The results showed





that most instruments provided reliable and consistent measurements, though discrepancies were noted for smaller particle sizes.

A functional process has been designed and validated to perform an ILC on APSD. It can be extended to ensure measurement accuracy across laboratories and paves the way for future ILCs to improve aerosol size distribution practices. The findings will contribute to better standardization, enhancing confidence in aerosol-related health assessments and regulatory applications.

**Author contribution**

GD and AK planned the ILC; CB, BDL, AN, RW and AK set-up the test bench and performed the measurements; GD analyzed the data and wrote the manuscript draft; GD, CB, BDL, AN, RW and AK reviewed and edited the manuscript.

**Competing interest**

The authors declare that they have no conflict of interest.

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
