# Peer review of "Interlaboratory comparison exercise for micro-aerosol size measurement by cascade impactor"

_Aerosol Research, 2024_

## Referee Comment (RC2)

In this paper, the authors presented an inter-comparison exercise for measuring micro-aerosol.

First of all, the scientific content of the article does not correspond to the quality of the journal. The study is not well written. Each section is detailed separately.

The introductory section does not sufficiently explain why it is important to look at each size range. This is missing from the introduction. The structure of the section is too general. The authors mention some previous studies but do not provide information on their results. This would be important for comparison.

In the Experimental Setup section, the authors do not provide detailed descriptions of each impactor. We should take into account the efficiency curves. What are the cut-off diameters? These are important parameters for comparison.

The authors write following:

Five aerosols with target aerodynamic diameters between 0.2 and 4 μm were selected for this ILC. Table 1 describes the characteristics of the solutions and generators used to produce these aerosols. The solutions are made of H2O, KCl and sodium fluorescein. Table 2 shows all the instruments used for the ILC. It should be noted that real-time monitoring of the particle size distribution is provided by an APS for aerosols larger than 1 μm.

The problem is that only particles larger than 1 μm can be monitored in real time for comparison.

In the analysis method section, they write about the different parameters, MMAD and geometric standard deviations. They write about the log-normal and Henry's method. How do they calculate? How they fit the data? Furthermore, references are missing.

In the result and discussion section, the author presented Z'-score and ζ-score calculated for MMAD and σg.

However, the novelty of the study, after the previous studies, is that it compares the results of cascade impactors (Andersen, DLPI, DLPI+) and Aerodynamic Particle Sizer. The fact that results for Participant 4 (APS) were only obtained for test A and B does not provide information on the wide range of particle sizes.

I do not consider the obtained results to be sufficient for publication. In addition, the whole text is full of typos, especially in the introduction. Please correct carefully.